# Evaluation of Parabens and Bisphenol A Concentration Levels in Wild Bat Guano Samples

**DOI:** 10.3390/ijerph20031928

**Published:** 2023-01-20

**Authors:** Slawomir Gonkowski, Julia Martín, Irene Aparicio, Juan Luis Santos, Esteban Alonso, Liliana Rytel

**Affiliations:** 1Department of Clinical Physiology, Faculty of Veterinary Medicine, University of Warmia and Mazury, Street Oczapowskiego 14, 10-719 Olsztyn, Poland; 2Departamento de Química Analítica, Universidad de Sevilla, C/Virgen de África, 7, E-41011 Sevilla, Spain; 3Department of Internal Diseases with Clinic, Faculty of Veterinary Medicine, University of Warmia and Mazury in Olsztyn, ul. Oczapowskiego 14, 10-719 Olsztyn, Poland

**Keywords:** wild animals, environmental pollution, endocrine disruptors, toxicology

## Abstract

Parabens and bisphenol A are synthetic compounds found in many everyday objects, including bottles, food containers, personal care products, cosmetics and medicines. These substances may penetrate the environment and living organisms, on which they have a negative impact. Till now, numerous studies have described parabens and BPA in humans, but knowledge about terrestrial wild mammals’ exposure to these compounds is very limited. Therefore, during this study, the most common concentration levels of BPA and parabens were selected (such as methyl paraben—MeP, ethyl paraben—EtP, propyl paraben—PrP and butyl paraben—BuP) and analyzed in guano samples collected in summer (nursery) colonies of greater mouse-eared bats (*Myotis myotis*) using liquid chromatography with the tandem mass spectrometry (LC-MS-MS) method. MeP has been found in all guano samples and its median concentration levels amounted to 39.6 ng/g. Other parabens were present in smaller number of samples (from 5% for BuP to 62.5% for EtP) and in lower concentrations. Median concentration levels of these substances achieved 0.95 ng/g, 1.45 ng/g and 15.56 ng/g for EtP, PrP and BuP, respectively. BPA concentration levels did not exceed the method quantification limit (5 ng/g dw) in any sample. The present study has shown that wild bats are exposed to parabens and BPA, and guano samples are a suitable matrix for studies on wild animal exposure to these substances.

## 1. Introduction

Currently, environmental pollution with organic compounds remains on the increase. Among these substances, bisphenol A (BPA) and parabens are of great importance [1,2]. BPA—chemical name 2,2-Bis(4-hydroxyphenyl)propane—is a substance used in production of plastics including polycarbonates and epoxy resins, which in turn is utilized in a wide range of branches of industry [2,3]. The popularity of BPA is because items containing this substance are light, resistant to damage and resilient [4]. It is not without significance that the synthesis of BPA is relatively easy and cheap [2,4]. Therefore, BPA is present in various everyday issues, such as bottles, electronic elements, furniture, food containers and many others [2,3,4].

In turn, parabens are a group of the esters of p-hydroxybenzoic acid and differ from each other by the type of substituent, which may be an alkyl chain or an aromatic ring [1]. Properties of parabens depend on the type of substituent in a molecule, but generally this group of components is characterized by fungistatic, anti-yest and anti-mold activities, as well as antibacterial properties [5,6]. These characteristics of parabens cause their widespread usage as preservatives in cosmetics, personal care products, drugs and food [1]. The presence of parabens has been described in shampoo, lipsticks, food packaging, baby wipes and many others, and the most used parabens are compounds with short-chain substituents, such as methyl paraben (MeP), ethyl paraben (EtP), propyl paraben (PrP) and butyl paraben (BuP) [1,5,6].

BPA and parabens, penetrating the body, bind to estrogen receptors and first of all cause disturbances in endocrine system [1,2]. Therefore, these substances are included in the group of compounds called endocrine disruptors. Both BPA and parabens may penetrate to the natural environment [6,7]. The presence of these substances has been described in the surface and tap water, soil, air, house dust and plants around the world [1,2,8,9,10]. Moreover, BPA and parabens penetrate the human and animal organisms, mainly through the gastrointestinal tract, but also through respiratory system, transdermal absorption and in utero through placenta [1,2,4]. Till now, BPA and parabens have been described mainly in the blood serum and urine [1,2,11,12,13]. Apart from these “classic” matrices, their presence has also been found in hair, nail, breast milk, semen and various tissues [1,2,4,14,15,16,17].

Endocrine disruptors can impair the functions of many internal organs. Previous studies have reported that BPA adversely affects the nervous system, endocrine glands, immune cells, male and female reproductive organs, heart and many other organs and internal systems [2,4,18]. It is also known that high exposure to BPA results in higher risk of diabetes, obesity, hypertension, cancers and neurodegenerative processes [2,19,20,21]. In turn, for many years, parabens were considered substances that had no negative impact on living organisms [1]. However, latest studies have shown that they have cytotoxic and genotoxic effects and negatively impact on the nervous, reproductive and endocrine systems, as well as immune reactions [1,5,22,23]. Moreover, some studies have described correlations between parabens and obesity, developmental disorders and neoplasms [22,24].

Contrary to humans, animal exposure to BPA and parabens polluting the environment is much less known. Regarding domestic animals, parabens have been found in the urine of dogs, cats and cows [25,26], raw cow’s milk [27] and canine fur [28]. More is known about the exposure of domestic animals to bisphenol A, the presence of which has been reported in the blood serum and urine of dogs and cats [29,30,31], canine fur [32], fresh pork meat [33] and cow’s milk [27].

Even less is known about the exposure of wild animals to BPA and parabens. Previous studies have reported the presence of BPA in wild water animals including mussels, fish, prawns and mollusks, as well as sea birds and seals, but knowledge about wild mammal exposure to BPA is rather scanty and fragmentary [34,35,36,37]. Parabens have also been described mainly in water animals—fish, birds and mammals [38,39,40,41,42]. Only single previous reports describe parabens in terrestrial birds and mammals [42].

Such a limited number of studies on wild mammals may be related to the fact that it is virtually impossible to collect samples of blood, urine or fur without capturing or killing the animal. The only matrix that can be taken from a wild animal for toxicology studies without significantly interfering with its life are feces/guano samples. The utility of this matrix is also supported by the fact that previous studies have confirmed that both BPA and parabens are excreted from organisms through the digestive tract [43,44].

On the other hand, among terrestrial mammals, bats play an important role in toxicological studies. Bats are an extremely heterogenous group of mammals consisting of above 1400 species living on all continents except Antarctica [45]. Bats are particularly vulnerable to the impact of factors changing their natural habitats, among which one of the most important is anthropogenic pollutions of the environment [46]. For this reason, bats are considered to be one of the best bioindicators of quality environment, including anthropogenic environmental pollution [46,47,48]. The high sensitivity of bats to environmental pollution is related to their biology, namely, their relatively long lifespan (even more than 30 years) and high metabolic rates resulting in high food intake, as well as frequent establishment of colonies near human habitation [49,50]. Unfortunately, despite the legal protection of most species, the number of bat populations is constantly decreasing [45], and one of the reasons for this is environmental pollution.

Taking these facts into account, the purpose of this study is to assess the exposure of the greater mouse-eared bat (*Myotis myotis*)—one of the most popular bat species of Poland [51] to bisphenol A and the most common parabens, such as MeP, EtP, PrP, BuP through guano samples analysis. According to the best knowledge of authors, it is the first study in which guano samples are used to evaluate the wild terrestrial mammal exposure to BPA and parabens, as well as the first ever study biomonitoring parabens in wild terrestrial mammals.

## 2. Materials and Methods

### 2.1. Reagents

The analytical reagents used in this study such as formic acid and ammonium acetate were purchased from Panreac (Barcelona, Spain). C18 disperser sorbent was provided by Scharlab (Barcelona, Spain). HPLC-grade methanol, and water were purchased from Romil (Barcelona, Spain). Paraben and BPA standards (≥99.0%) were purchased from Sigma-Aldrich (Steinheim, Germany). The internal standards (ISs) BPA-d14 and PrP-^13^C_6_ were supplied by Cambridge Isotope Laboratories (Tewksbury, MA, USA). Individual stock standard solutions of 1000 mg/L and working solutions (by dilution of the former) were prepared in methanol.

### 2.2. Sample Collection

Four summer (nursery) colonies of greater mouse-eared bats (*Myotis myotis*) located in various parts of Poland were included into the study. The localization and characteristics of colonies are presented on Figure 1. The choice of bat colonies was not accidental. The colonies included into the study are some of the largest colonies of a greater mouse-eared bat in Poland. They are located in various environments and in various parts of Poland: colony no. 1 is located in a small village, which is placed relatively close to the most industrialized region in Poland (Upper Silesia); colony no 2 is located in a small village in agricultural land without industry; colony no 3 is located in a medium-sized town with a chemical industry and colony no 4 is located in a small town without industry. Such different localization of bat colonies allowed to better understand, what factors may influence of bat exposure to parabens and BPA.

Guano samples were collected in August–September 2021. To collect samples, flat glass containers were put for 48 h on the floor of the places with bat colonies. After these time containers were removed and samples were transferred to glass jars and frozen at −20 °C until further analysis. Forty guano samples (ten samples from each colony) collected from different parts of rooms, where bat colonies live, were included into the present study. During collection of samples, special emphasis was placed on not stressing and scaring the bats. Because the sampling procedure was completely non-invasive, according to Act for the Protection of Animals for Scientific or Educational Purposes of 15 January 2015 (Official Gazette 2015, No. 266), applicable in the Republic of Poland, consent of bioethical committee to conduct the present study was not required.

### 2.3. Sample Treatment

Samples were lyophilized and homogenized. An aliquot of 1.0 g of sample was fortified with 100 µL of a methanol solution of the ISs (250 ng/mL) in 12 mL glass tubes. The sample procedure involves the extraction of the analytes by ultrasonic solvent extraction, using 7 mL of methanol (0.5% *v*/*v*, formic acid) as extraction solvent for 5 min and centrifuged for 5 min (4050× *g*). The extraction procedure was repeated three times and the supernatants were combined. Given the complexity of the selected biological sample and of the extract obtained in the first stage of ultrasonic solvent extraction (USE), the cleaning process was intended to remove any interference from the extract. For this approach, a cleaning process using dispersive adsorbents (d-SPE) based on the QuEChERs technique (Quick, Easy, Cheap, Effective, Rugged and Safe) was used. This procedure consisted in shaking vigorously for 2 min the combined extracts from USE with 0.3 g of C18 sorbent and then centrifuged at 4050× *g* for 5 min. Finally, the supernatant was evaporated to dryness, reconstituted in 0.25 mL of a mixture methanol–water (50:50 *v*/*v*) and filtered through a 0.22 µm nylon filter prior to injection into the liquid chromatography tandem mass spectrometry (LC-MS/MS) instrument (Agilent, Santa Clara, CA, USA). Chromatographic conditions were those previously reported by Martín et al. [52]. Instrument settings and analytical determination parameters are summarized in the Appendix A).

### 2.4. Method Validation, Quality Assurance and Quality Control

The analytical features (linearity, sensitivity and accuracy (trueness and precision)) of the method are presented in Table 1.

A matrix-matched calibration curve method was used to overcome the matrix effect. For that, fortified commercial guano samples were prepared containing the analytes at eight different concentration levels in the range from method quantitation limit (MQL) to 100 ng/g dw. Note that the commercial guano samples used for matrix-matched calibration were simultaneously analyzed and their signals were subtracted to spiked sample extract signals. BPA and parabens (except MeP) were not detected in the commercial guano samples.

Method detection limits (MDLs) and method quantification limits (MQLs) were calculated as the concentrations of each compound corresponding to a signal-to-noise ratio of 3:1 and 10:1, respectively, using guano spiked samples at low concentration levels.

Accuracy (trueness and precision) of the method was assessed using commercial guano samples spiked at 25 ng/g dw. Accuracy was evaluated by a recovery control over the whole procedure, which included extraction from the matrix, d-SPE and concentration step. The precision is expressed through the relative standard deviation (%, RSD) of measurements on different days.

To guarantee the quality assurance of results, a protocol involving the use of control spiked samples (fortified commercial guano samples at 50 ng/g dw), solvent (methanol:water 50:50 *v*/*v*) injections, standards containing a mixture of the target compounds (12.5 ng/mL) and procedural blanks (processed in the same way as the samples) were included into each analytical batch (5 samples).

### 2.5. Statistical Analysis

The statistical analysis of the differences in concentration levels of substances studied between particular bat colonies was made up with GraphPad Prism version 9.2.0 (GraphPad Software, San Diego, CA USA) and a Kruskal–Wallis test was used. The differences were considered statistically significant at *p* < 0.05.

## 3. Results

During the present investigation, parabens were present in guano samples collected in all bat colonies (Table 2).

Among parabens, MeP was a substance that was quantified in all samples analyzed in concentration levels. Its levels fluctuated from 14.00 ng/g dry weight (dw.) to 142.00 ng/g dw. The second most common paraben was EtP, which had concentration levels higher than MQL, was observed in 62.5% of guano samples, and its concentration levels fluctuated from <0.05 ng/g dw to 239 ng/g dw. In turn, PrP in concentration levels above MQL was noted in 42.4% of samples with a range from <0.05 ng/g dw. to 229 ng/g dw. The least common paraben in the samples included into the study was BuP. The concentration levels of this paraben higher than the method detection limit (MDL) were noted only in two guano samples (5% of all samples). In one of them BuP concentration levels achieved 3.61 ng/g dw, and in the second 27.5 ng/g dw.

Contrary to parabens, BPA concentration levels in the vast majority of samples did not exceed MDL. Only in one sample the value was slightly higher but did not exceed MQL (Table 2 and Table 3).

During the present study, clear differences in parabens concentration levels between particular animals in one colony were noted. The most visible such differences were noted in bat colony no. 1 (Table 2). Moreover, statistically significant differences in MeP concentration levels were noted between colonies. The highest mean concentration of this paraben was found in colony no. 3, where this value was statistically significantly higher than values noted in colonies no. 1 and no. 4 (Table 4). Differences in concentration levels of other parabens between colonies were not statistically significant (Table 4).

Moreover, differences in the frequency of occurrence of EtP, PrP and BuP in concentrations higher than MQL were noted between colonies. The highest frequency was noted in colony no. 1, and the lowest in colony no. 4 (Table 4).

## 4. Discussion

The results obtained during the present investigation have shown that parabens are present in guano samples, which indicates the exposure of wild bats to these substances. MeP, a paraben with the shortest functional group, has been noted in the highest concentration levels in all samples studied. In turn, other parabens studied in this investigation have been found less frequently and in lower concentration levels. Generally, it is in agreement with the previous investigations, which have reported that MeP is the paraben that is most commonly and abundantly found in the natural environment, including water, soil and air [8,53,54], as well as in humans [55,56] and wild marine animals [39,40,41,42].

Moreover, in the present study differences in parabens concentration levels in guano samples between particular bat colonies were clearly visible. It is in agreement with previous observations concerning both human [5,6,8,54] and wild animal exposure (Table 5) to parabens, which unequivocally confirm that paraben levels noted in the living organisms depend to a large extent on the place, where the study is performed.

It relates to the human origin pollution of the natural environment, human population density, industrialization, as well as the frequency of using cosmetics, personal care products and medicines by the inhabitants of a given area [1,5,6]. Moreover, clear differences in parabens concentration levels in particular guano samples collected in the same colony were also noted in the present study. This situation has been also observed in previous studies on parabens levels in wild animals (Table 5) and in humans living in the same area. For example, the levels of MeP in the human urine samples collected in Queensland (Australia) ranged from 74.4 to 1180 ng/mL [55], and samples collected in Seville (Spain) achieved from 68.3 to 14,187 ng/g [56]. The present results strongly confirm that local factors may influence on the parabens levels in particular individuals and also in wild bats.

The comparison of present results with previous investigations is difficult, because in spite of the fact that paraben levels have been determined in some species of wild animals (Table 5), till now, according to the best knowledge of authors, feces/guano samples were not used at all in this type of research. Moreover, the majority of previous studies concern marine animals (fish, birds or mammals), and only very limited observations have been conducted on terrestrial animals (Table 5). In addition previous studies have been conducted in in completely different parts of the world than the present research.

For these reasons, it is difficult to relate the results concerning MeP concentration levels in bat guano to data from previous literature (Table 5). On the other hand, levels of other parabens observed in bat guano samples are higher than those noted in the majority of previous studies on wild animals, in which these parabens were usually not detected (Table 5). Due to such a high correlation between the study site and exposure to parabens, it seems advisable to analyze the levels of parabens in Poland to better understand the obtained results. It should be pointed out that knowledge about this issue is relatively limited in comparison to studies in other parts of the world. It is known that parabens are present in surface water in Poland. In lake water, MeP concentration level amounted from 1.7 to even 1578 ng/L [63]. The levels of other parabens are clearly lower and achieved values 0.8–27.5 ng/L for EtP, 0.5–93.9 ng/L for PrP and 0.6–22.6 ng/L for BuP [63]. Much higher levels of parabens have been observed in wastewater, where limits of MeP, EtP, PrP and BuP achieved 2235.0–40,898.6 ng/L, 791.2–8169.4 ng/L, 542.2–7803.3 ng/L and 68.8–710.7 ng/L, respectively [64]. Moreover, parabens have also been found in the fur of dogs living in Poland. In this case mean concentration levels of MeP, EtP and PrP achieved 176 ng/g, 48.4 ng/g and 79.8 ng/g, respectively [28].

Despite other matrices used in previous studies and the comparative difficulties resulting from this, it can be concluded that the parabens levels in wild bat guano in Poland are generally lower than the levels of these compounds in humans and domestic animals [28,65,66,67]. This is logical due to the fact that humans create these environmental pollutants and are exposed to them through the use of cosmetics, personal care products, medicines and food containers [1,5,6]. In turn, companion animals, which live in close proximity to humans, have direct contact with the owners and are exposed to similar factors as humans [28]. Unlike humans and domestic animals, bats that are exposed to parabens, which can be found in the environment such as in water, air or food, tend to have a lower degree of exposure, even if their summer colonies are located near human populations.

In addition to parabens, the presence of BPA in bat guano samples was also found in the present study. However, BPA concentration levels were lower than MQL in all samples investigated. To date, only one study has described bat exposure to BPA. In that study, BPA in the median concentration level of 397 ng/g was described in bat carcasses collected in the northeastern United States [68]. Therefore, BPA concentration levels noted in the present study are clearly lower. Other previous studies (more numerous than studies on parabens) have described BPA in various aquatic animals including fish, mussels, prawns, mollusks, as well as seabirds [34,35,36,37].

Similarly to parabens, the degree of exposure to BPA depends greatly on where the study is conducted [2,3,4]. Therefore, to evaluate the degree of bat exposure noted in the present study, it is important to show results of earlier research on BPA in the living organisms and environment in Poland (Table 6).

It should be pointed out that feces/guano samples have extremely rarely been used to analyze the exposure to BPA. Such studies have been conducted on feces collected from Baltic seals, but these animals were not wild, but kept in the Marine Station in Hel (Poland) belonging to the Institute of Oceanography of the University of Gdansk [36]. These studies have shown the presence of BPA in the seals’ feces with a concentration ranging from 20.06 ng/g to 75,659.78 ng/g, i.e., much higher than results obtained in the present study [36]. A similar situation occurred in the case of wild seabird guano, where the concentration of BPA ranged from 41.85 ng/g to 2701.9 ng/g [36,77]. Comparing the previous data with the current results, it can be concluded that BPA concentration levels in bat guano are significantly lower than levels of this substance both in the feces samples, as well as other matrices collected from humans, domestic and aquatic animal species in Poland (Table 6).

It should be underlined that the unequivocal explanation of the sources of parabens and BPA in bat guano, as well as the differences in paraben concentration levels between particular bat colonies, is rather difficult without comprehensive studies of the environment, where the colonies are located for the presence of parabens and BPA in the surface water, air, soil, and insects, which are the food of bats. It can only be supposed that bat exposure to parabens and BPA is mainly connected with contamination of surface water and bat food with these substances. Such conclusion is supported by the fact that the presence of parabens and BPA in the surface water in various parts of the world is commonly known [2,8,9,10,63]. Regarding the food of bats, the matter is less clear. The greater mouse-eared bats are insectivorous and till now, the exposure of land insects to parabens and BPA has not been studied. On the other hand, the common presence of these substances in the soil and water [8,9,10,63,81,82,83,84] strongly suggests that also the body of beetles—the main food of the greater mouse-eared bats—may contain these substances. The third major source of bat exposure to parabens and BPA is likely to be the air. Many previous studies have reported parabens and BPA in the indoor air and house dust, what is connected with the presence of these substances in the building materials, paints, varnishes, epoxy resins, furniture and other items used by the inhabitants [85,86,87,88]. The fact that bat colonies included into the study are located in buildings used by people strongly suggests that the indoor air and dust may be the important source of bat exposure to parabens and BPA.

Differences in paraben levels noted between particular bat colonies may result both from local factors including disparities in the buildings, where colonies are located (different construction and finishing materials, different purpose of the building), but confirmation of this thesis requires thorough environmental studies. Differences may also result from general environmental pollution and industrialization. The highest MeP levels have been noted in the colony no. 3 located in Pulawy, which is the largest town among the towns included in the study. Additionally, the chemical industry company is located in Pulawy. In turn, higher levels of EtP, PrP and BuP (although without statistical significance) have been noted in colony no. 1, located in Brenna. Brenna is a rather small village, but it is located relatively close to Upper Silesia—the most industrialized region in Poland and Ostrawa—which is a large industrial center in the Czech Republic. This fact seems to affect the bat exposure to parabens, because lowest parabens concentration levels have been noted in colony no. 2 in Sliwice—a small village situated on agricultural land and colony no. 4 in Opole Lubelskie—a small town without industrial centers.

The question arises whether fecal samples are a suitable matrix for studying the exposure of wild animals to parabens and BPA. Previous experimental studies have shown that after oral and transdermal administration of parabens, not only a large percentage of these substances (about 4%) was excreted through the gastrointestinal tract, but the majority of parabens was extracted through urine [44]. However, the present results, which have shown the presence of parabens in bat guano samples, strongly suggest that this matrix is suitable to perform paraben biomonitoring in wild animals. The situation is slightly different with BPA. In spite of the fact that BPA common in the natural environment and living organisms (Table 6), the concentration levels of this substance noted in present study were relatively low and did not exceed the level of MQL. On the other hand, previous studies on farm animals, including cows, pigs and chickens [89,90], as well as the aforementioned studies on seals and seabirds [36,77], have proven that feces/guano samples can be used to assess animal exposure to BPA. Moreover, it is known that BPA levels in feces reflect the degree of exposure to BPA administered by both the oral route and (to a lesser extent) transdermal adsorption [43]. Therefore, the results obtained in the present study in which BPA concentration levels were significantly lower than those noted in previous studies (Table 6) may conversely indicate a relatively low exposure of wild bats in Poland to BPA, and on the other hand—a reduced level of BPA in guano resulting from the metabolism of this substance in bats. Due to the fact that till now there is no information about BPA metabolism in bats, the exact elucidation of this issue requires further comprehensive studies.

The next important question that arose during the current research concerns the possibility to determine of correlations between the content of parabens and BPA in the guano and the dose of these compounds to which bats are exposed. Based on the data obtained in the present study, this is difficult, because such determinations require thorough studies of the content of studied compounds in individual elements of the environment. Moreover, metabolism of endocrine disruptors and thus the main route of their elimination from the body may be different in different species of mammals [2,22,91,92]. As mentioned above, parabens and BPA metabolism in bats has not been studied so far. Therefore, determination of doses of these substances to which bats are exposed may be only hypothetical and based on experimental studies conducted on other animal species. However, considering such experimental studies [43,44,93,94], it can be assumed that doses of parabens and BPA to which bats are exposed may be from a few to even several dozen times higher than the concentration of these substances in the guano samples. In the light of previous experimental studies conducted on other mammal species, such doses of parabens and BPA, due to their endocrine disrupting properties may, influence on the reproductive, nervous and endocrine systems, as well as on the immune cells [95,96,97,98,99]. However, it must be remembered that natural environment significantly differs from experimental conditions. In the laboratory, the experiment status health of animals is rather good, and their age and condition is usually equalized. Moreover, usually only one toxic factor is studied. In the case of animals in the natural environment, their condition and health may be very different, and often numerous toxic substances with synergistic action may affect the animal organism. Therefore, although studies of the toxicity of parabens and BPA in bats have not been performed so far, it can be assumed that in this species the exposure to endocrine disruptors has a similar effect to that noted in experimental investigations.

However, despite the necessity of further research in order to explain the aforementioned doubts, guano/feces samples seem to be a good alternative to “classic” matrices in studies on the exposure of wild animals to parabens and BPA. Feces sample collection, contrary to collection of urine and blood samples, is completely stress-free and does not involve the need to capture the animal, which is particularly important in the case of protected animals.

## 5. Conclusions

The present study has described for the first time the presence of parabens and BPA in guano samples of the wild bats, which indicates the exposure of wild animals to these substances, confirming previous observations regarding the widespread distribution of parabens and BPA in the natural environment. In the present study, the highest concentration levels were noted in the case of MeP, which is an agreement with previous investigations that describe this substance as the most common paraben in the surface water, air and living organisms. On the other hand, the levels BPA noted in the present study were clearly lower than those noted in the feces of seals and guano of seabirds, which may suggest that bats are less exposed to BPA than aquatic species. In the light of present study, it can be assumed that parabens and BPA, as endocrine disruptors, may influence on the status of wild bat health.

Current research also confirms the usefulness of feces/guano samples (as an easily obtained material) for studies on the degree of exposure of wild animals to parabens and BPA, which is of particular importance in protected animals. However, due to intraspecies differences between parabens and BPA metabolism and toxicity and body distribution, further research is needed to evaluate all aspects connected with wild bat exposure to endocrine disruptors which pollute the environment, as well as the use of feces/guano samples for this type of research.

## Figures and Tables

**Figure 1 ijerph-20-01928-f001:**
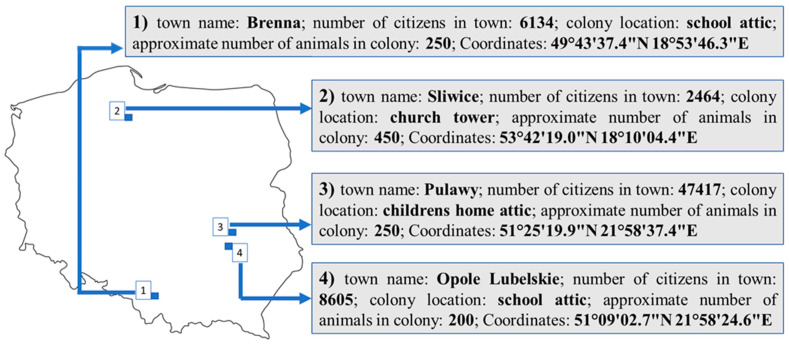
Location and characteristics of bat colonies included into the study.

**Table 1 ijerph-20-01928-t001:** Linearity, method quantification limit, precision and recovery of parabens and BPA in guano matrix.

Compound	Linearity	MDL	MQL	RSD	Rec
	R^2^	(ng/g dw)	(ng/g dw)	(%)	(%)
MeP	0.998	0.02	0.05	12.1	98.6
EtP	0.998	0.02	0.05	11.2	108.1
PrP	0.999	0.02	0.05	14.2	107.2
BuP	0.998	0.02	0.05	9.5	118.8
BPA	0.997	2.00	5.00	35.9	139.3

MDL: Method detection limit MQL: Method quantification limit; RSD: Relative Standard Deviation; Rec: Recovery.

**Table 2 ijerph-20-01928-t002:** Concentration levels (ng/g dw) of parabens and bisphenol A in bat guano samples.

Bat Colony No.	Sample No.	Concentration (ng/g)				
		MeP	EtP	PrP	BuP	BPA
1	1	31.1	0.68	0.41	<MDL	<MDL
	2	25.5	0.65	0.30	<MDL	<MDL
	3	27.8	0.75	0.48	<MDL	<MDL
	4	20.7	0.23	0.08	<MDL	<MDL
	5	52.7	35.3	18.6	<MDL	<MDL
	6	25.1	1.48	0.81	<MDL	<MDL
	7	25.6	1.89	2.09	<MDL	<MDL
	8	56.1	55.3	46.4	3.61	<MDL
	9	142	239	229	27.5	<MQL
	10	26.9	3.48	4.38	<MDL	<MDL
2	1	33.6	2.27	2.59	<MDL	<MDL
	2	50.3	1.73	1.51	<MDL	<MDL
	3	60.4	2.38	1.59	<MDL	<MDL
	4	43.8	0.95	0.99	<MDL	<MDL
	5	35.9	0.60	0.42	<MDL	<MDL
	6	40.6	0.41	<MQL	<MDL	<MDL
	7	38.6	0.55	<MQL	<MDL	<MDL
	8	35.9	0.46	<MQL	<MDL	<MDL
	9	37.2	0.67	<MQL	<MDL	<MDL
	10	43.7	<MQL	<MQL	<MDL	<MDL
3	1	70.2	1.60	1.33	<MDL	<MDL
	2	72.7	0.92	<MQL	<MDL	<MDL
	3	46.6	<MQL	<MQL	<MDL	<MDL
	4	80.3	0.84	<MQL	<MDL	<MDL
	5	73.5	1.07	<MQL	<MDL	<MDL
	6	96.4	<MQL	<MQL	<MDL	<MDL
	7	86.4	<MQL	<MQL	<MDL	<MDL
	8	100	1.28	1.34	<MDL	<MDL
	9	71.2	<MQL	<MQL	<MDL	<MDL
	10	119	0.70	<MQL	<MDL	<MDL
4	1	50.8	<MQL	<MQL	<MDL	<MDL
	2	24.4	<MQL	<MQL	<MDL	<MDL
	3	14.0	<MQL	<MQL	<MDL	<MDL
	4	22.6	<MQL	<MQL	<MDL	<MDL
	5	22.9	<MQL	<MQL	<MDL	<MDL
	6	24.2	<MQL	<MQL	<MDL	<MDL
	7	24.4	<MQL	<MQL	<MDL	<MDL
	8	25.4	<MQL	<MQL	<MDL	<MDL
	9	45.3	<MQL	<MQL	<MDL	<MDL
	10	24.3	<MQL	<MQL	<MDL	<MDL

Compound acronyms: MeP: methylparaben; EtP: ethylparaben; PrP: Propylparaben; BuP: butylparaben; BPA: bisphenol A; <MQL: Below Method Quantification Limit EtP = 0.05 ng/g dw; PrP = 0.05 ng/g dw.; BPA = 5 ng/g dw.); <MDL: Below Method Detection Limit (BuP = 0.02 ng/g dw.; BPA = 2.00 ng/g dw.).

**Table 3 ijerph-20-01928-t003:** (ng/g dw.) and frequency of detection of parabens and bisphenol A in the analyzed guano samples (n = 40)—cumulative data.

Compound	Range (ng/g)	Arithmetic Mean	Geometric Mean	Median	Frequency of Detection above MQL
MeP	14.00–142.00	48.70	41.81	39.6	100
EtP	<0.05–239	14.21	1.61	0.95	62.5
PrP	<0.05–229	18.37	1.78	1.45	42.5
BuP	<0.02–27.5	15.56	9.96	15.56	5
BPA	<2.00–<5.00	-	-	-	0

Compound acronyms: MeP: methylparaben; EtP: ethylparaben; PrP: Propylparaben; BuP: butylparaben; BPA: bisphenol A; MQL—Method Quantification Limit.

**Table 4 ijerph-20-01928-t004:** Concentration values (ng/g dw.) and frequency of detection of parabens and bisphenol A in the analyzed guano samples taking into account the place of collection.

Bat Colony Number	1	2	3	4
**MeP**				
Number of samples, in which levels were higher than MQL	10	10	10	10
Minimum	20.70	33.60	46.60	14.00
Maximum	142	60.40	119	50.80
Mean ± SD	43.35 ± 36.7 ^A^	42.00 ± 8.13	81.63 ± 19.97 ^AB^	27.83 ± 11.21 ^B^
**EtP**				
Number of samples in which levels were higher than MQL	10	9	6	0
Minimum	0.23	0.41	0.7	-
Maximum	239	1.97	1.6	-
Mean ± SD	33.88 ± 74.51	1.11 ± 0.79	1.07 ± 0.33	-
**PrP**				
Number of samples in which levels were higher than MQL	10	5	2	0
Minimum	0.08	0.42	1.33	-
Maximum	229	2.17	1.34	-
Mean ± SD	30.26 ± 71.34	1.42 ± 0.8	1335 ± 0.01	-
**BuP**				
Number of samples in which levels were higher than MDL	2	0	0	0
Minimum	3.61	-	-	-
Maximum	27.5	-	-	-
Mean ± SD	15.56 ± 16.89	-	-	-
**BPA**				
Number of samples in which levels were higher than MQL	0	0	0	0
Minimum	-	-	-	-
Maximum	-	-	-	-
Mean ± SD	-	-	-	-

Statistically significant differences ^A^—between bat colony 1 and 3 (*p* = 0.0084), ^B^—between bat colony 3 and 4 (*p* < 0.0001). Compound acronyms: MeP: methylparaben; EtP: ethylparaben; PrP: Propylparaben; BuP: butylparaben; BPA: bisphenol A; MQL—Method Quantification Limit; MDL—Method Detection Limit.

**Table 5 ijerph-20-01928-t005:** Selected previous studies on paraben concentration levels in wild animals. Paraben concentration levels are shown in ng/g (in solid matrices) or ng/mL (in liquid matrices).

Species	Localization	Matrix	n	Concentration Levels				Ref.
				MeP	EtP	PrP	BuP	
Dolphin (various species)	Gulf of Mexico, Florida (USA)	liver	17	<41.1–865	n.d.	<2.05–3.47	n.d.	[39]
		blubber	17	n.a.–<8.21	n.d.	n.d.	n.d.	
Pygmy sperm whale	Atlantic ocean/Anciote river	liver	3	<20.5–37.7	n.d.	<4.10	n.d.	
		blubber	3	n.a.	n.d.	n.d.	n.d.	
Sea otter	California Coast (USA)	liver	25	n.a.–126	<4.1–11.8	<1.03–3.95	n.d.	
		kidney	10	n.a.–360	n.d.	n.d.	n.d.	
		brain	10	5.99–77.2	<1.03–3.26	n.d.	n.d.	
	Washington Coast (USA)	liver	18	<20.5–358	n.d.	n.d.	<4.1–31.8	
Northern sea otter	Alaskan Coast (USA)	liver	14	n.a.–686	<8.21–31.6	n.d.	n.d.	
Polar bear	Alaska (USA)	liver	10	<4.10–16.9	n.d.	n.d.	n.d.	
Common dolphin	Korean coastal waters	blubber	6	1.3–7.9				[40]
		muscle	6	13–121				
		melon	6	4.8–12				
		stomach	6	44–118				
		liver	6	13–224				
		testis/ ovary	6	26–76				
		brain	5	4.2–50				
		uterus	2	37–74				
Finless porpoises	Korean coastal waters	blubber	6	6.4–21				
		muscle	6	12–88				
		melon	6	21–71				
		stomach	6	75–228				
		kidney	6	181–359				
		liver	6	235–569				
		testis/ovary	6	44.3–256				
		stomach content	2	59–76				
Fish (various species)	Greater Pittsburgh Area (USA)	brain	58	n.d.	n.d.	n.d.	n.d.	[41]
Fish (various species	Taihu Lake (China)	muscle	199	88.1–1200	33.6–450	55.3–543	<LOQ–40.0	[57]
Fish and bivalves (various species)	Atlantic ocean, pacific Ocean, Mediterranean Sea, Victoria lake (Uganda)	body	64	0.8–32	n.d.	n.d.	n.d.	[58]
Fish (various species)	New York State (USA)	muscle	50	n.a.−690	n.d.	n.d.	n.d.	[42]
Fish (various species)	Florida (USA)	liver	6	<2.01–44.3	n.d.	n.d.	n.d.	
		muscle	6	n.a.–43.9	n.d.	n.d.	n.d.	
		kidney	1	18.8	n.d.	n.d.	n.d.	
		gill	1	71	n.d.	n.d.	n.d.	
		brain	1	735	n.d.	n.d.	n.d.	
Black bear	Michigan (USA)	liver	2	33.5–58.2	n.d.	n.d.	n.d.	
		kidney	2	24.0–37.6	n.d.	n.d.	n.d.	
Sea eagle	Baltic Sea area	liver	20	<8.01–657	n.d.	n.d.	n.d.	
Albatrosses (various species)	Sand Island, Midway (USA)	liver	15	<8.01–23.4	n.d.	n.d.	n.d.	
		kidney	16	<8.01–18.2	n.d.	n.d.	n.d.	
		brain	12	<1.99–6.66	n.d.	n.d.	n.d.	
		fat	10	n.a.	n.d.	n.d.	n.d.	
		muscle	12	<2.01–15.7	n.d.	n.d.	n.d.	
		eggs	13	n.a.	n.d.	n.d.	n.d.	
Bald eagle	Michigan (USA)	liver	1	796	n.d.	n.d.	n.d.	
		plasma	15	<0.2–0.37	n.d.	n.d.	n.d.	
		kidney	1	580	n.d.	n.d.	n.d.	
		muscle	2	87.4–169	n.d.	n.d.	n.d.	
Herring gull	Michigan (USA)	eggs	9	<1.99–14.0	n.d.	n.d.	n.d.	
Common cormorant	Michigan USA)	eggs	4	n.a.	n.d.	n.d.	n.d.	
Loon	Carteret County (USA)	liver	10	<8.01–336	n.d.	n.d.	n.d.	
Loon	Maine and New Hampshire (USA)	eggs	15	<1.99–22.2	n.d.	n.d.	n.d.	
Fish (various species)	Yangtze river (China)	plasma	36	11.6–39.5	0–10.5	0–6.97	0–b1.46	[38]
Fish (various species)	Yangtze river (China)	bile	35	8.17–21.9	0–31.6	2.19–112	0–4.42	[59]
Fish (various species)	Spain/various rivers	muscle	59	n.d.–84.69	n.d.–0.82	n.d.–7.43		[60]
Striped Catfish	Colombia/various rivers	muscle	20	<20–32.0				[61]
Clams	Antarctica	body	7	<2.1–5.8	n.d.	n.d.–5.3	n.d.	[62]
Sea urchin		body	1	5.7	n.d.	n.d.	n.d.	
Fish (Emerald rockcod)		body	7	5.1–26.9	n.d.	n.d	n.d.	
		liver	1	2.4	n.d.	n.d.	n.d.	

MeP: methylparaben; EtP: ethylparaben; PrP: Propylparaben; BuP: butylparaben; n.a.—not available; n.d.—not detected; Ref.—references.

**Table 6 ijerph-20-01928-t006:** Previous studies on BPA concentration levels (in ng/g for solid matrices and ng/mL in liquid matrices) in living organisms and environment in Poland.

Matrix	n	BPA Concentration	References
Human hair	42	26.1–1498.6	[34]
	20	3.6–52.9	[69]
Human blood serum	155	0.11–112.1	[70]
	245	0.05–4.017	[71]
	40	4.4–55.1	[72]
	52	4.3–55.3	[73]
Human breast milk	20	0.09–11.56	[74]
Human urine	250	0.05–53.1	[75]
Seal blood serum	108	<LOQ–81.58	[76]
Seal milk	44	0.1–406	[76]
Seal fur	17	<LOQ–137.2	[34]
Seal feces	60	20.06–75,659.78	[36]
Dog fur	30	<MDL–436	[32]
Pig muscles	5	13.77–49.86	[33]
Water bird whole body	10	19.7–440.1	[77]
Water bird feather	26	29.3–512.4	[34]
Water bird intestine	52	6.6–1176.2	[78]
Water bird blood	53	<0.07–30.6	[78]
Water bird lungs	53	<2.0–331.7	[78]
Water bird guano	7	41.6–2701.9	[77]
	30	41.85–1666.54	[36]
Sea fish muscles	12	25.4–798.4	[77]
Sea mussels	10	6.8–197.2	[77]
	148	Nd–273.6	[79]
Sea zooplankton	5	105.7–769.2	[77]
Sea water	120	<LOQ–0.2779	[80]
	71	0.00377–0.82271	[36]
Inland surface water	105	NQ–0.095	[63]
Sea sediment	72	0.08–26.39	[36]
Beach sand	71	0.83–15.80	[36]

LOQ—limit of quantification; MDL—method detection limit; NQ—not quantified.

## Data Availability

Data are available upon request.

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
