# Peer review of "Evaluation of Parabens and Bisphenol A Concentration Levels in Wild Bat Guano Samples"

_ijerph, 2023, doi:10.3390/ijerph20031928_

Round 1

Reviewer 1 Report

The authors conducted an interesting study investigating wild bats’ exposure to parabens and BPA in Poland, by collecting the feces in their natural habitat. The study is novel and quite meaningful. However, some method development and validation (MDMV) details are lacking and the discussion is missing a lot of key points regarding the rationale of this study. Major comments are as follows:

1.      This study is a detection study. Because of this, MDMV is very important. Rather than simply describing how QAQC was done, please also add a method validation section as the first part of the result, and describe how MDL and MQL are calculated as well as list MDL and MQL for each analyte (how was blank contamination treated especially in the case of BPA), linearity, spike recovery, etc. It actually surprised me to see that BPA is undetectable in the fecal samples, even with such a low MQL of 5 ng/g dw.

2.      The whole discussion needs rewriting. The authors struggled with little previous literature on paraben level in wildlife to compare to, and many related BPA literature to compare to while their samples have undetectable BPA. A quick search gave me some reports on paraben levels in wildlife. Please conduct a better literature review and reflect it in the introduction and discussion.

·        Elevated Accumulation of Parabens and their Metabolites in Marine Mammals from the United States Coastal Waters. Jingchuan Xue, Nozomi Sasaki, Madhavan Elangovan, Guthrie Diamond, and Kurunthachalam Kannan. Environmental Science & Technology 2015 49 (20), 12071-12079. DOI: 10.1021/acs.est.5b03601

·        Tissue-Specific Accumulation and Body Burden of Parabens and Their Metabolites in Small Cetaceans. Yunsun Jeong, Jingchuan Xue, Kyum Joon Park, Kurunthachalam Kannan, and Hyo-Bang Moon. Environmental Science & Technology 2019 53 (1), 475-481. DOI: 10.1021/acs.est.8b04670

·        Renz, L., Volz, C., Michanowicz, D. et al. A study of parabens and bisphenol A in surface water and fish brain tissue from the Greater Pittsburgh Area. Ecotoxicology 22, 632–641 (2013). https://doi.org/10.1007/s10646-013-1054-0

3.      Please think about whether it makes sense to compare paraben and BPA levels in wildlife to those in humans. This study is investigating how much wild bats are exposed to these hazardous chemicals by monitoring their feces, with a further goal of trying to understand what potential damage such exposure can do to wild bats. We humans created these environmental pollutants and we are very likely to have higher exposure to these chemicals than wildlife. The standpoint of this work is environmental toxicology rather than human-centered public health.

4.      Discussion on toxicity and the potential consequences of such a level of exposure is very limited. Parabens and BPA are endocrine disruptors. The author only listed “endocrine disruptor” in the keywords but mentioned nothing about this in the whole manuscript. This part should be a focus in the discussion. A couple of potential discussion points here:

·        External exposure: What is causing the differences in the four sampling locations? Considering paraben and BPA levels in the environment (soil, water (like lines 253-259), plants (potentially the food of the bats)), what is likely to be the major source of exposure to these chemicals?

·        Internal exposure: Can we know the internal exposure level from the fecal paraben and BPA level? Comparing to other literature reporting the ADME of parabens in wildlife, especially mammals, might provide some insights. How does this level of exposure compare to the toxicity level? What is likely to be the consequence of such levels of exposure? I understand that toxicology studies on bats may be scarce, but there are plenty of in vitro and experimental animal tox studies, as well as some studies on wildlife. The authors can generate a very good discussion based on this.

5.      Minor comments as below:

Lines 136-138: please delete the repetitive sentence.

Lines 138-139: please describe the SPE procedure – was the samples simply pass through the C18 cartridge or there was an elution using specific solvents? If so, please describe the whole process.

Lines 142-143: please give the brand name and the model of the LC-MSMS system used

Lines 148: how to make sure that commercial guano samples do not contain target analytes? If target analytes are actually in it, how to deal with it?

Lines 221-227: there is no need to discuss interindividual variability.

Reviewer 2 Report

We congratulate the authors for the manuscript.

The study presented relevant and worrying data on exposure to parabens and BPA in wild animals. Studies have focused on the human species, which makes this study extremely relevant.

We suggest to the authors that in the discussion they better explore the characteristics of the sampled regions, since there was a difference in the concentration of the compounds between the groups of bats. Furthermore, it is suggested that the choice of sampling points be explained in the methodology.

Round 2

Reviewer 1 Report

Thank you for the revision. The revised manuscript looks better. I only have a couple of minor comments:

1.       Please move Table S2 into the main text. This table is very important.

2.       Lines 57-61: Please move the first sentence to the beginning of the previous paragraph. Readers need to understand the toxicity of BPA and parabens first and realize why we care about their environmental occurrences and wildlife exposure. Remove “Due to the fact that estrogen receptors are widespread in living organisms”. BPA also has anti-androgenic activity.

3.       Lines 83-85: merge this into the previous paragraph. Remove “The knowledge about parabens levels in wild animals is also relatively scanty…” Your previous paragraph started with this sentence.

4.       Line 156: need to define “USE” in line 151 “ultrasonic solvent extraction”.

5.       Line 163: change “solvent” to “supernatant”. I assume it is the supernatant collected from the previous centrifugation step that goes through evaporation and reconstitution.

6.       Line 180: remove “The method was in-house validated.”

Author Response

The authors thank for positive review. All suggestions of the Reviewer have been taken into account. All changes have been marked by red font.

  1. Table S1 has been included in the main text of the manuscript (in new version table 1 – line 163
  2. The introduction has been reedited according to the suggestions of the Reviewer (49-51 and line 60)
  3. The paragraphs has been reedited according to the suggestions of the Reviewer (lines 79-81)
  4. Term “USE” has been define (line 148)
  5. The term “solvent” has been replaced by “supernatant” (line 153)
  6. Change has been made according to the suggestion of the Reviewer (line 161).